# Revealing single-neuron and network-activity interaction by combining high-density microelectrode array and optogenetics

Toki Kobayashi [1,4] ✉, Kenta Shimba [2,4] ✉, Taiyo Narumi[2], Takahiro Asahina [3], Kiyoshi Kotani[2] & Yasuhiko Jimbo [1]

The synchronous activity of neuronal networks is considered crucial for brain function. However, the interaction between single-neuron activity and network-wide activity remains poorly understood. This study explored this interaction within cultured networks of rat cortical neurons. Employing a combination of high-density microelectrode array recording and optogenetic stimulation, we established an experimental setup enabling simultaneous recording and stimulation at a precise single-neuron level that can be scaled to the level of the whole network. Leveraging our system, we identified a network burst-dependent response change in single neurons, providing a possible mechanism for the network-burst-dependent loss of information within the network and consequent cognitive impairment during epileptic seizures. Additionally, we directly recorded a leader neuron initiating a spontaneous network burst and characterized its firing properties, indicating that the bursting activity of hub neurons in the brain can initiate network-wide activity. Our study offers valuable insights into brain networks characterized by a combination of bottom-up self-organization and top-down regulation.

Neuronal networks in the brain often exhibit spontaneous synchronous activity, which is believed to play a crucial role in information processing, storage, and transmission[1–4], as well as in neurological conditions including epileptic seizures[5,6]. Synchronization represents the macroscopic state of a network that emerges from the propagation of electrical activity through the synaptic connections of neurons. Understanding the interaction between this macroscopic state and single neurons within a network is essential for elucidating the mechanisms of the brain, where neurons are the fundamental building blocks.

However, the interaction between the activity of single neurons and network activity remains poorly understood. Specifically, the changes in the properties of single neurons depending on their synchronous state are not well documented. The brain is known to adapt its cognitive processes based on synchronous rhythms[1,7]. Excessive synchrony, as seen in epileptic seizures, causes cognitive impairment[8–10]. Thus, the synchronous state can affect the dynamics of individual neurons. However, few studies have explored how the synchronization state affects the activity of single neurons. Extrapolating to general networks, the question arises of

[1]Department of Precision Engineering, School of Engineering, The University of Tokyo, Tokyo, Japan. [2]Department of Human and Engineered Environmental Studies, Graduate School of Frontier Sciences, The University of Tokyo, Chiba, Japan. [3]Center for Information and Neural Networks, National Institute of Information and Communications Technology, Osaka, Japan. [4]These authors contributed equally: Toki Kobayashi, Kenta Shimba. ✉ e-mail: kobayashi.toki.jb@gmail.com; shimba@neuron.t.u-tokyo.ac.jp

how macroscopic network states affect the individual nodes of the network.

How a single neuron generates network-wide activity remains poorly understood. Previous studies on brain slice connectivity[11–13] have shown networks with a scale-free topology, highlighting the presence of hub neurons with many connections. Simulation and experimental studies have indicated that these hub neurons contribute significantly to the emergence of network activity[12,14,15]. However, the specific involvement of hub neurons in network activity is not well elucidated, particularly regarding their role in initiating network activity versus simply propagating it from various starting points. Similar inquiries have arisen in research on social media and infectious diseases, e.g., does the dissemination of misinformation start from a social media influencer?[16] Is the super spreader in an infectious explosion a social hub?[17–19]

To clarify these interactions, an experimental system that can analyze whole neuronal networks at the single-neuron resolution is essential. Cultured neuronal networks composed of dissociated neurons offer a valuable platform for gaining fundamental insights into both single-neuron and network synchronization[14,20]. These networks represent the simplest form of neuron connectivity and exhibit network activity in the form of network bursts and scale-free dynamics[21,22]. Through the study of cultured neuronal networks at single-neuron resolution, we can shed light on the interaction between network-wide activity and individual neurons. Specifically, our investigation aimed to elucidate 1) how network burst affects single-neuron properties and 2) the involvement of hub neurons in initiating network burst.

In an experimental system aimed at studying this interaction, several components are essential:

(i) activity recording of neuronal networks at single-neuron resolution.
(ii) Activity control through flexible, targeted stimulation at single-cell resolution.
(iii) Long-term experiments to examine changes in activity, connectivity, and network states of individual neurons.

While optical imaging enables the simultaneous recording of a large number of single neurons, long-term recording is hindered by phototoxicity[23,24]. The patch-clamp method generally allows the electrical recording and stimulation of individual neurons, whereas conducting network recording and long-term experiments is difficult due to experimental complexity and invasiveness, respectively.

Extracellular recording using microelectrode arrays (MEAs) has been used for noninvasive long-term recording of neuronal networks at multiple points[25–27]. Recently, high-density microelectrode arrays (HD-MEAs) based on CMOS technology have provided simultaneous long-term recordings at single-neuron resolution in neuronal networks[28–30]. HD-MEAs allow for the control of neuronal activity through electrical stimulation at each electrode. However, recording the activity of neurons immediately after stimulation is impeded by artifacts, and achieving single-neuron stimulation is challenging due to the simultaneous activation of surrounding neurons.

To overcome these limitations, a combination of optogenetic stimulation and HD-MEA recordings would be suitable. Optogenetic stimulation is a well-established method for controlling the activity of single neurons[31–33]. Previous studies have constructed experimental systems combining conventional MEA recording and optogenetic stimulation[34–41]. Three stimulation methods were employed for optogenetic stimulation. First, whole-area stimulation, which is unsuitable for generating single-neuron activity, was used to generate a network-wide response[35,40]. The second method leveraged an integrated device with optical stimulation channels and recording electrodes[37]. Although some studies have achieved single-neuron stimulation[42], owing to the fixed position of the stimulation channel, it is impossible to flexibly stimulate targeted single-neurons by adopting the location in the culture device. The third method employed a digital mirror device (DMD) or spatial light modulator to generate a light pattern for optical stimulation[36,39,43]. This method can flexibly stimulate targeted neurons with a single-neuron resolution. However, no established experimental system can optically stimulate single neurons in HD-MEA setups.

Here, we constructed an experimental system that combines electrical recording via HD-MEA with optogenetic stimulation employing a DMD. Our objective was to elucidate how network activity affects single neurons' properties and how specific neurons with a hub role initiate network bursting. Our system enabled the generation of consistent direct responses in single neurons as well as indirect synaptic responses. Through our experimental system, we elucidated the latency changes in synaptic transmission during network activity. Additionally, we experimentally validated that the bursting activity of a hub neuron triggers network-wide activity.

## Results

### Combining electrical recording and optogenetic stimulation

For realizing stimulation and recording of neuronal network with single-cell resolution, an experimental setup that integrates electrical recording via HD-MEA and optogenetic stimulation using a DMD (Fig. 1a). The HD-MEA recording unit was positioned beneath an upright microscope, which was connected to the DMD. An opaque black acrylic box shielded the experimental system from external light sources. To ensure the long-term survival of the neurons, the box was maintained at 37 °C using a heater, and the cells were supplied with $CO_2$ via air from a $CO_2$ incubator (refer to Fig. 1b).

For the preparation of cultured neuronal networks, rat cortical neurons were cultured on HD-MEAs containing 26,400 electrodes with a 17.5-µm pitch. To facilitate optogenetic stimulation, ChR2-GFP was introduced into the cells using an adeno-associated virus (AAV). The expression of ChR2-GFP was confirmed through fluorescence microscopy (Fig. 1c, d), and mature neuronal networks ( > 30 days in vitro)[21,27] were used in the experiments that followed.

### Optical stimulation for reliable direct response and synaptic response

Considering the short-term plasticity of synaptic transmission, we hypothesized that the latency of information transfer between individual neurons would vary depending on network activity. To test this hypothesis, it is imperative to consistently elicit responses in individual neurons through light stimulation and identify neurons that respond indirectly by transmitting information.

To evaluate the ability to control the activity of a single neuron, we optically stimulated neurons and recorded their responses. To determine the stimulation areas, fluorescence images of GFP were taken (Fig. 1d, step 1). The stimulation area was divided into grids and superimposed with the GFP fluorescence image (Fig. 1d, step 2). Grids containing ChR2-GFP-expressing neurons were manually selected (Fig. 1d, step 3). The optical stimulus was delivered to each selected location sequentially (Fig. 1e). To ensure the reliable induction of spikes in a single neuron only once during the stimulation period, we applied a 50 × 50 µm² optical stimulation with a 5 ms pulse at an intensity of 15.4 mW/mm², which was determined based on our pilot studies (Supplementary Fig. 1).

The direct response spikes of the stimulated neurons were detectable, albeit with stimulation artifacts present on the stimulated electrodes (see Fig. 2a). To evaluate these artifacts, we stimulated HD-MEAs containing only the culture medium. The waveform of the stimulation artifact varied depending on the stimulation intensity and duration (refer to Supplementary Fig. 2). The amplitude of the artifacts was smaller than the spike amplitude for the 50 × 50 µm² stimulation area. Artifacts were mitigated using bandpass filtering (300–3500 Hz). Narrowing the stimulation area was critical because stimulating the entire electrode generated an artifact of several hundred µV, rendering

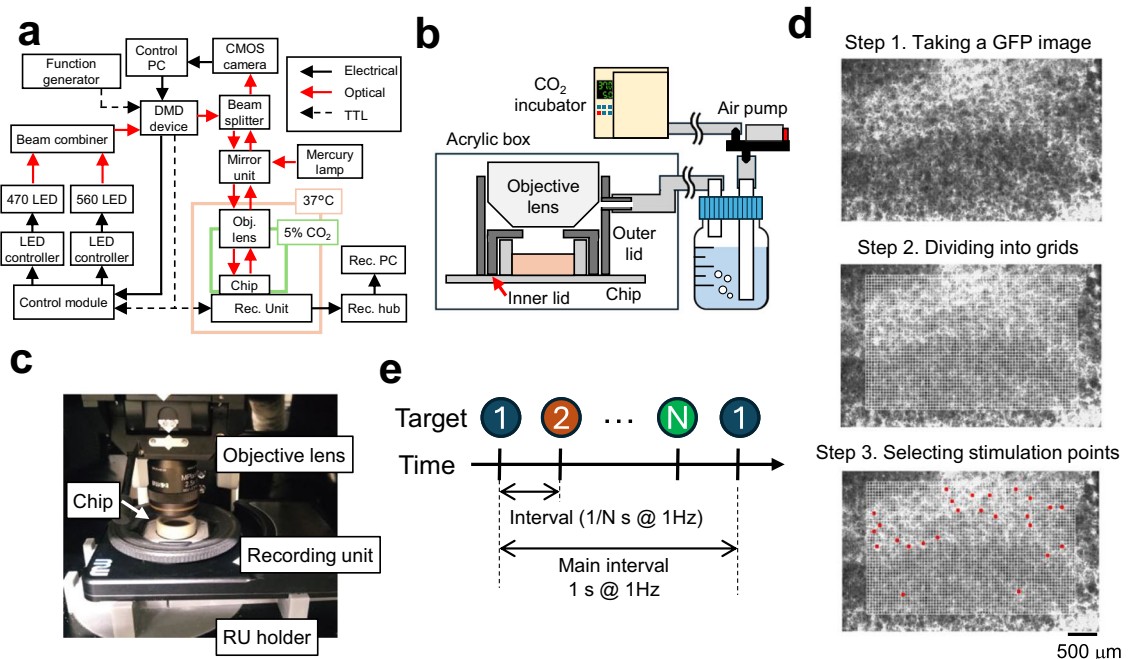

**Fig. 1 | The experimental setup that combines simulation using a digital mirror device with recording via a high-density microelectrode array. a** Overview of experimental system, including electrical, optical, and TTL signal flow. **b** System for supplying humidified $CO_2$-containing air. **c** A photo of the recording system. **d** Procedure for selecting stimulation locations. A GFP fluorescence image was divided into grids. Stimulation locations were manually selected (red squares). **e** Optical stimulation procedure. Optical stimuli were delivered to each stimulation location sequentially.

direct response measurement impossible (refer to Supplementary Fig. 2).

Our experimental system allowed spikes to be consistently induced in the targeted single neurons with minimal jitter during the stimulation period (see Fig. 2a). Additionally, minimal jitter responses of 79 neurons out of the 122 stimulated neurons were simultaneously recorded from the network (see Fig. 2b) with high reproducibility (Fig. 2c). Employing these optical stimulation parameters, we stimulated 321 neurons with a stimulation frequency of 1 Hz at each neuron across the four HD-MEA chips for 1 hour. We recorded reliable direct responses ( > 95%) in 244 neurons (77.3%) during the stimulation period (4.43 ± 1.21 ms) with low jitter (standard deviation: 0.62 ± 0.48 ms).

Synaptically evoked indirect responses have also been observed. Figure 3a displays the post-stimulus potentials based on the spatial arrangement of the electrodes. Direct responses were recorded at the electrode under the optical stimulus (see Figs. 3b, 1). However, at the electrode situated 70 μm away from the one depicted in Figs. 3b, 1, indirect responses with considerable jitter were recorded after the stimulus period (see Figs. 3b, 2). This jittered activity probably originates from indirectly responding neurons, which receive activity propagated via synaptic connections from the directly responding neurons. In two independent samples, we stimulated 66 locations and detected 66 directly responding neurons and 125 indirectly responding neurons. The responding neurons and optical stimulation locations are shown in Supplementary Fig. 3. This is consistent with findings from previous studies[44,45]. The delays and high jitter in indirect responses can be attributed to delays in axon conduction ( ~ 200 μm/ms), synaptic transmission ( ~ 3 ms), and dendrite integration[44,46].

A previous study reported that electrical stimulation with the frequency of 1 Hz induced long-term depression (LTD)[47]. When LTD is induced, characterizing neurons that respond via synaptic connections becomes difficult because synaptic transmission efficacy decreases. Thus, we evaluated the response latency and probability (Supplementary Fig. 4). Response latency decreased and response probability increased for 5-10 min after the stimulus was initiated. The same trend was observed afterwards, but the change was less than 5%.

These results indicate that the stimulation may have caused a kind of plasticity, such as synaptic plasticity or facilitation, although no LTD was induced. However, because response probability was maintained at a high level ( > 0.95), the change in response latency was small ( < 5%), and the slow change compared to the change associated with the network burst, this plasticity does not affect the following results of this study.

In addition to targeted single-neuron stimulation, our system can stimulate a combination of single neurons. For proof-of-concept, two locations were jointly stimulated, and the response probability at each recording electrode was compared with that of the single-site stimulation of each (Supplementary Fig. 5a, 5b). For the combined stimulation, some electrodes showed a greater change in response probability than when the responses of multiple single stimuli were added together (Supplementary Fig. 5b). From the difference between aggregate response and response to combined stimulation, we found both electrodes with increased and decreased activity with the combined stimulation ( > 20% increase, 8.1% and 5.8%; <20% decrease, 1.5% and 1.2%; two different electrodes settings).

To demonstrate the capability of recording single-neuron activity, electrical activity was overlaid on immunofluorescence images. We observed directly responding neurons followed by indirectly responding neurons, indicating synaptically responsive neurons (see Fig. 3c, d, and Supplementary Movie 1). Thus, we successfully induced and recorded the activity of directly responding neurons, fired directly by optical stimulation, and indirectly responding neurons, fired due to synaptic connections.

Our experimental system can record multiple neurons simultaneously without any neuron selection biases, making it suitable for studying synaptic integration. As shown in Fig. 4, three different neurons directly responded to a specific stimulus (see Fig. 4a), while one neuron consistently showed indirect responses to three different stimuli. The amplitude landscape (see Fig. 4b) and average waveform (see Fig. 4c) of the neuron, indicated by a green circle, exhibited similar shapes. In addition, the potential map of the electrical activity showed that the direction of electrical activity transmission via the axon

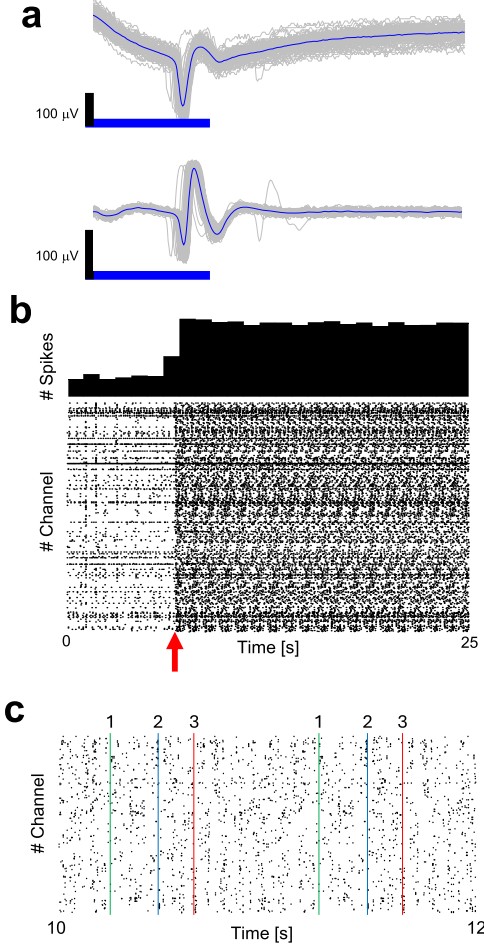

**Fig. 2 | Electrical activity of neurons evoked by optical stimulation. a** Example of post-stimulus signal. Top: raw signal. Bottom: filtered signal (300–3500 Hz band-pass). Gray: signal of each stimulus. Blue: stimulus time-averaged signal. **b** Top: histogram of number of spikes for optical stimulation of 122 cells at 1 Hz. Bin width is 1 s. Bottom: raster plot for all 1024 recording channels. The red arrow indicates the onset of optical stimulation. **c** Magnified raster plot. Representative stimulation times are indicated with green, blue and red vertical lines. Because the stimulation frequency for each neuron was 1 s optical stimulus was applied to the same location at a 1-s interval.

remained consistent among the three types of stimuli (Supplementary Movie 2). These results suggest that the spikes originated from the same neuron. Of the indirectly responding neurons, 4% were neurons integrating two stimuli, and 0.8% were neurons responding to three stimuli (*n* = 130 neurons). We identified integrator neurons that received synaptic inputs from multiple neurons.

### Burst-dependent response change of single neurons

The latency between the response and stimulus reflects single-cell properties such as membrane potential and ion channel states. Changes in latency show alterations in individual neuron properties. To elucidate the effects of network activity on a single neuron, latency was monitored for 1 hour.

Figure 5a shows burst-dependent changes in response latency to optical stimuli at a representative neuron, with the timing of network bursts indicated. Neuronal networks under periodic optical stimulation showed occasional network bursts (identified by red vertical lines in Fig. 5a). Immediately after the network burst ceased, the latency changed discontinuously and then recovered continuously (Fig. 5a, b, and c). Moreover, substantial changes in response times persisted in indirectly responding neurons (refer to Fig. 5d).

Subsequently, the burst-dependent changes were quantified (see Methods). First, the coefficient of variation (CV) of response latency was calculated for evaluating the variance over time. Network burst globally increased CV of the response latency in the network (Fig. 5e). Then, to evaluate the reproducibility of burst-dependent changes, burst-dependent response change (BDRC) was calculated as the time difference between the response latency after a network burst and the average of 10 latencies before the burst. Neurons exhibiting BDRC were sorted in order of highest mean BDRC, and all BDRC values were plotted (Fig. 5f). The top 36% of neurons (26 out of 73 neurons) showed a prolonged mean latency of >1 ms, with 92.7% of BDRC events being prolonged latencies. On the other hand, only 5.5% of neurons (4 out of 73 neurons) showed a decrease in BDRC, which tended to have a large standard deviation. These results suggest that network bursts generally prolonged response latency and increase the variance.

Instances where the end of one burst and the start of the next burst were <2 s apart were grouped as successive bursts. Successive bursts induced larger and longer BDRCs compared to single bursts (refer to Fig. 5g and h). These findings suggest that neurons adapt their responses to optical stimulation, and network bursts disrupt this adaptation. Successive bursts cause large and lasting changes in the response patterns of single neurons. In summary, network activity causes changes in the activity patterns of individual neurons.

### A leader neuron initiates network bursts

Finally, we examined how specific neurons with hub roles initiate a network burst. As shown in Fig. 6a, b, and Supplementary Movie 3, 1.3% of neurons (4 locations out of 290 stimulation locations from eight independent samples) initiated network bursts when stimulated with a pulse of 0.2 Hz, with a network burst initiation probability of 10%, 18%, 60%, and 98%. During stimulation-induced network bursts, electrical activity was consistently transmitted with a fixed pattern starting from the leader neuron (Ind. in Fig. 6c, d), which is similar to the propagation pattern of spontaneous bursts (Sp.1 in Fig. 6c, d). Thus, the leader neuron exhibited effective connectivity with all neurons in the network. Through immunostaining after recording, we successfully identified leader neurons (Fig. 6e).

Importantly, the leader neuron-initiated network bursts during spontaneous activity (Sp.1 in Fig. 6d). When the leader neuron manifested burst firing, the network bursts were initiated with a probability of 100%. Additionally, network bursts originating from other locations were observed (Sp.2 in Fig. 6d). Among the network bursts during spontaneous activity, 33% originated from the leader neuron, while 67% originated from other locations. The spike waveforms recorded near the leader neuron were similar, indicating that the source of electrical activity was a single neuron (refer to Supplementary Fig. 6). These results provide experimental evidence that spontaneous network bursts are initiated by a single neuron.

To characterize the properties of the leader neuron, we examined functional connections among the network and its inter-spike interval (ISI). The leader neuron was classified as the information sender, and network activity was generated from a cluster of sender neurons, including the leader neuron (Supplementary Fig. 7). As shown in Fig. 6f, the histogram of ISI was bimodal. Neurons in the cerebral cortex are classified based on their firing patterns in intracellular recordings, such as regular spiking, fast-spiking, and intrinsically bursting[48–50]. During spontaneous activity, the leader neuron fired at approximately 100 Hz with an interval of approximately 8 s (refer to Fig. 6f). Thus, the leader is considered an intrinsically bursting neuron. These results suggest the presence of intrinsically bursting neurons in the cortex that initiate network-wide activity.

To further characterize the burst generating local circuits, optical stimuli were applied to the target locations and 50 μm

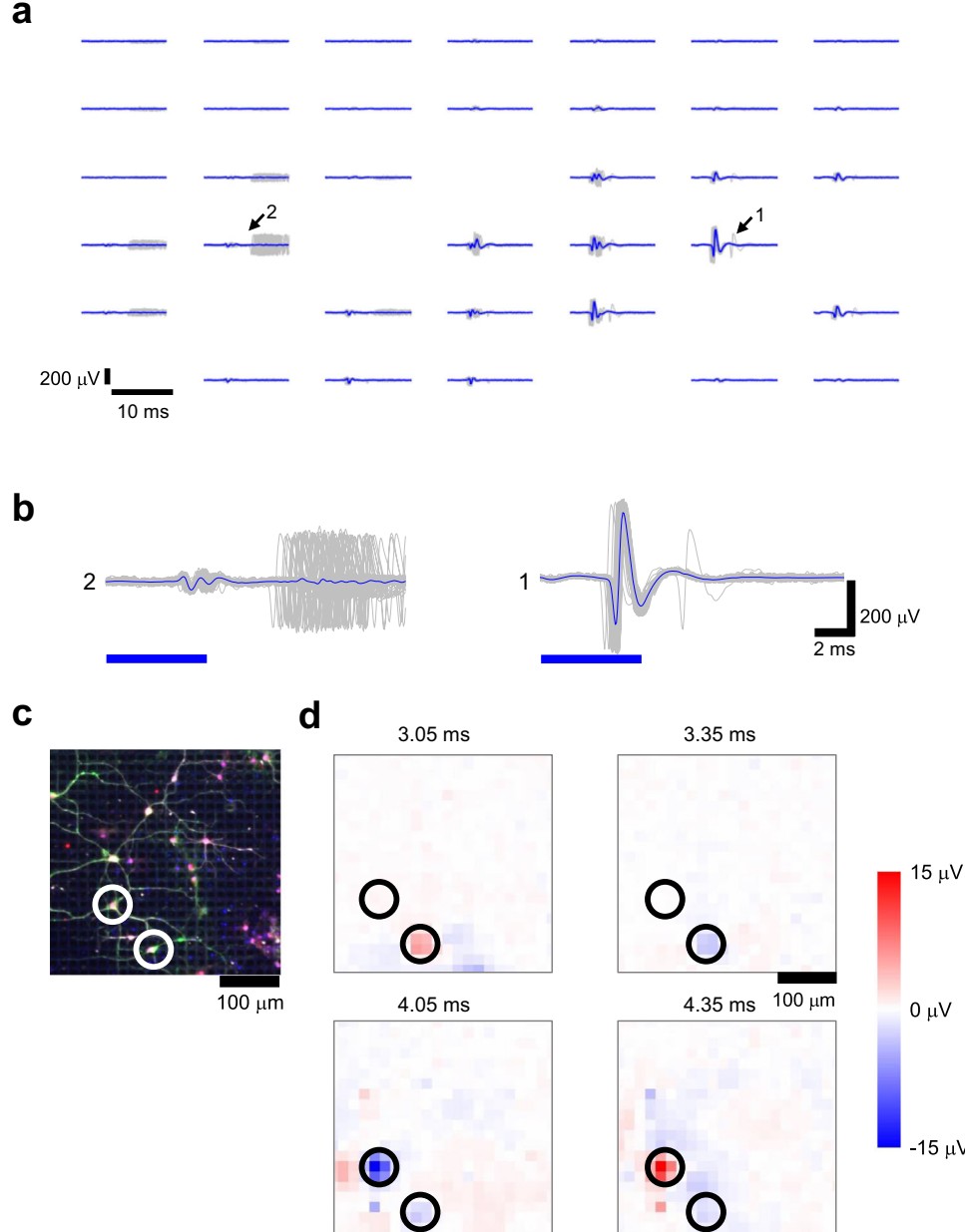

**Fig. 3 | Recording and activating synaptic evoked indirect responses of single neurons. a** Spatial distribution of stimulus-time-triggered signal. Each signal was recorded at the electrode at each location (17.5 μm pitch). Gray: signal for each stimulus. Blue: stimulus-time-triggered averaged signals. **b** Signals of electrodes marked with arrows and numbers. **b-1**: direct responses. **b-2**: indirect responses. **c** Confocal fluorescence image of immunostained neurons. Soma and dendrite of neurons, gray (MAP2); soma of neurons, red (NeuN); GFP-expressing neurons, green; cell nucleus. blue (DAPI). Immunofluorescence staining was repeated using two independent samples with similar results. **d** Post-stimulus extracellular potential heatmap.

shifted locations (Supplementary Fig. 8a), for a total of five locations. Results showed that bursts were sometimes induced by stimulation of nearby locations (Supplementary Fig. 8b). Pattern analysis showed that bursts induced by stimuli in close proximity showed a similar propagation pattern (Supplementary Fig. 8cd). On the other hand, the electrode order of firing tended to be different for the electrodes near the stimulus (Supplementary Fig. 8c). Latency to network bursts was calculated and found to be significantly different for each stimulus location (Supplementary Fig. 8ef). These results suggest that local neural circuits are involved in the generation of network bursts.

Overall, these results demonstrate the feasibility of our experimental system for investigating how a single node and the entire network interact to generate network-wide activity.

## Discussion

In this study, we constructed an experimental system to bridge the gap between single-neuron activity and network-wide activity. Optogenetic stimulation with single-neuron resolution induced activity in individual neurons, while HD-MEA simultaneously recorded activity at single-neuron resolution in the whole network. We elucidated how network activity alters the properties of single neurons and demonstrated that a single neuron can initiate network bursting spontaneously. Our findings offer fresh insights into the interplay between single neurons and neuronal networks.

Previous experimental systems combining MEA recordings and optogenetic stimulation have not enabled recording and stimulation at single-neuron resolution[34,39–41]. This limitation arises from two main factors: (i) the optical stimulation area was not narrowed down to the

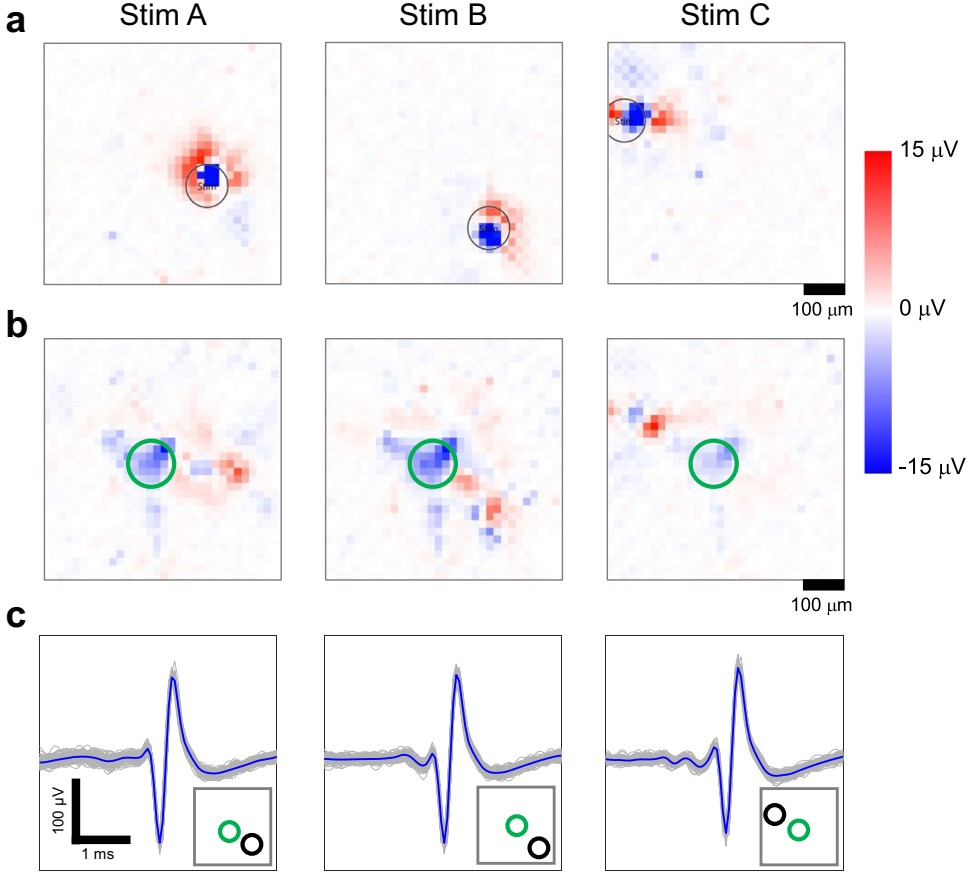

**Fig. 4 | Integrator neuron responding synaptically to three different positions of optical stimuli.** Heatmaps and spike signals of post-stimulus potential for three different stimuli. **a** Heatmaps of the spike time of a neuron that directly responded. The position of each optical stimulus is indicated by a black circle. The background is a GFP fluorescence image. **b** Heatmaps of spike time of an integrator neuron. The integrator neuron is indicated by a green circle. **c** Spike-time-triggered signals of the integrator neuron. Gray: signals for each stimulus. Blue: spike-time-triggered averaged signals. In the lower right-hand schematic, the location of the optical stimulus is indicated by a black circle, and the location of the integrator neuron is indicated by a green circle.

scale of a single-neuron resolution but instead encompassed the entire cultured neural network; (ii) the recording electrode interval of the MEA was large (>200 μm), precluding single-neuron scale (approximately 20 μm) recording. In our study, we achieved stimulation and recording at the resolution of a single neuron by leveraging the high spatial resolution of the DMD and HD-MEA (with a 17.5-μm electrode interval). This method has significantly contributed to various electrophysiological studies[51], including this study.

While network activity has been suggested to switch neuronal network states, few studies have experimentally elucidated the effects of state changes at the single-neuron scale. Here, we identified BDRC, a phenomenon wherein the response mode of single neurons switches depending on network bursts. Thus, network activity appears to modulate the activity of individual neurons. BDRC provides a single-neuron-scale explanation for network-burst-dependent response pattern changes in cultured neuronal networks. Therefore, BDRC is considered the mechanism underlying the network-burst-dependent loss of stimulus-specific information embedded in the network[52,53].

BDRC can be conceptualized as burst-dependent short-term plasticity, and multiple potential physiological mechanisms underlie BDRC[44,46,54]. The mechanisms in directly responded neurons include changes in ion permeability of ChR2 and/or other ion channels. In contrast, the mechanisms of BDRC in synaptically responsive neurons may include synaptic transmitter depletion in presynaptic cells, changes in the properties of synaptic transmitter receptors, and changes in ion channel permeability. Consequently, BDRC may be linked to mechanisms associated with epilepsy. Neuronal networks

experiencing BDRC may be interpreted as being unable to generate a typical response to external stimuli. Epileptiform firing, characterized by excessive synchronous activity in the brains of individuals with epilepsy, often results in transient cognitive impairment (TCI)[8,10]. Thus, it is hypothesized that BDRC occurs in epileptic brains, incapacitating a normal response to external stimuli and the manifestation of TCI.

Previous studies have demonstrated that neurons exhibit network-wide activity when stimulated[34,55,56], both in brain slices and cultured neuronal networks. In addition, it was shown that the electrodes reproducibly record spikes at the beginning of a spontaneous activity, suggesting that a single neuron or a small number of neurons may initiate a network burst in cultured neuronal networks[57,58]. Pasquale et al. recorded network activity with conventional MEA with 64 electrodes and showed that leader electrodes were rapidly recruited within both spontaneous and electrically induced bursts[59]. However, it remains unclear whether a single neuron can spontaneously and intrinsically initiate network activity across the entire network. We directly recorded and characterized a leader neuron that initiates spontaneous network bursts. Leader neurons exhibit intrinsic bursting properties, which is consistent with our previous observation that intrinsically bursting neurons trigger network bursts in response to electrical stimulation[55]. Notably, the leader neuron identified in this study bursts upon stimulation and spontaneously initiates network bursts. Additionally, because the leader neuron could be identified with immunostaining after recording, detailed analysis of protein or gene expression patterns can also be performed to further characterize the leader neuron. Leader neurons serve as effective hub

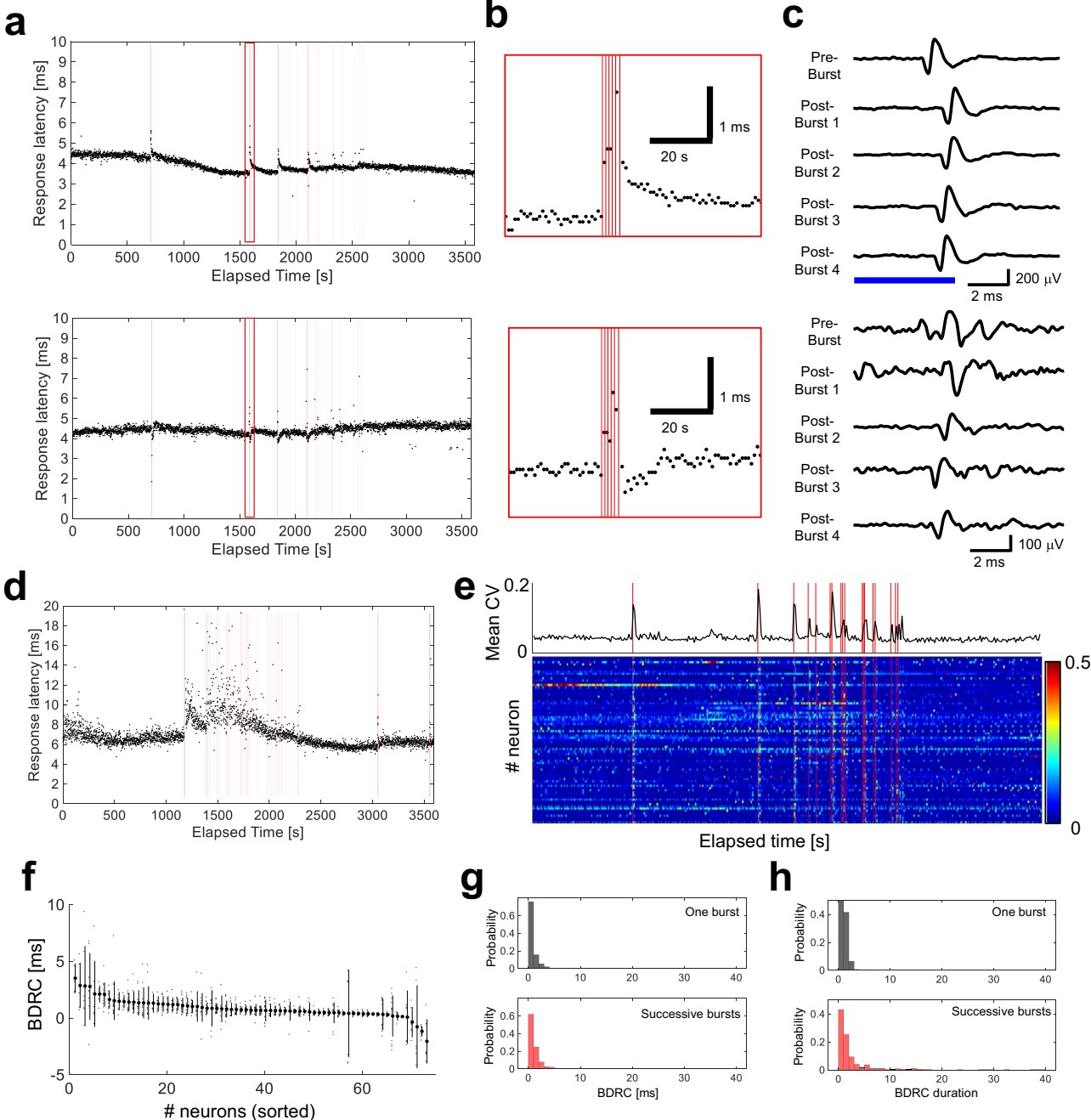

**Fig. 5 | Burst-dependent response change. a** Burst-dependent change in response latency to optical stimuli at a representative electrode. The red vertical lines indicate the time of network bursts. Top, prolonged latency; bottom, shortened latency. **b** Enlargements of **a**. **c** extracellular potential waveforms triggered at stimulus before and after network bursts in **b**. Pre-Burst indicates response to stimulus immediately before the network burst. Post-Bursts 1, 2, 3, and 4 indicate response to 1st, 2nd, 3rd, and 4th responses immediately after the network burst. The horizontal blue bar represents the stimulus duration (5 ms). **d** An example of the long-lasting change in response latency after a successive network burst.

**e** Burst-dependent change in variance of response latency. Coefficient of variation (CV) was computed at each neuron. Top, mean CV over all neurons; bottom, heatmap for each neuron. Note that network burst globally increased CV of response latency in the network. **f** Sorted burst-dependent response change (BDRC) for each neuron. BDRC was calculated as the time difference between the response latency after a network burst and the average of 10 latencies before the burst. Each dot indicates each BDRC value. Mean ± standard deviation.
**g** Comparison of BDRC between one and successive bursts. **h** Comparison of BDRC durations between one and successive bursts.

neurons that may contribute to the initiation of epileptic seizures in the brain, aligning with other simulation studies[15,60]. Understanding the properties of leader neurons and developing neuromodulation techniques to enhance or suppress their activity holds promise for neurological therapies.

Our experimental method offers several advantages: (1) versatility, (2) high-throughput parallel data acquisition, (3) flexible stimulation, and (4) a low computational cost for extracting single-neuron

activity. First, our method can be applied to any culture that is compatible with HD-MEA and capable of introducing light-sensitive proteins. Our method was successfully applied to sensory neurons and directly recorded saltatory conduction in axons[51]. Additionally, our method can be employed to induce and record the activity of iPSCs, organoids, and slices on an HD-MEA[30,61].

Second, our method facilitates high-throughput parallel stimulation and the recording of single-neuron activity. We successfully

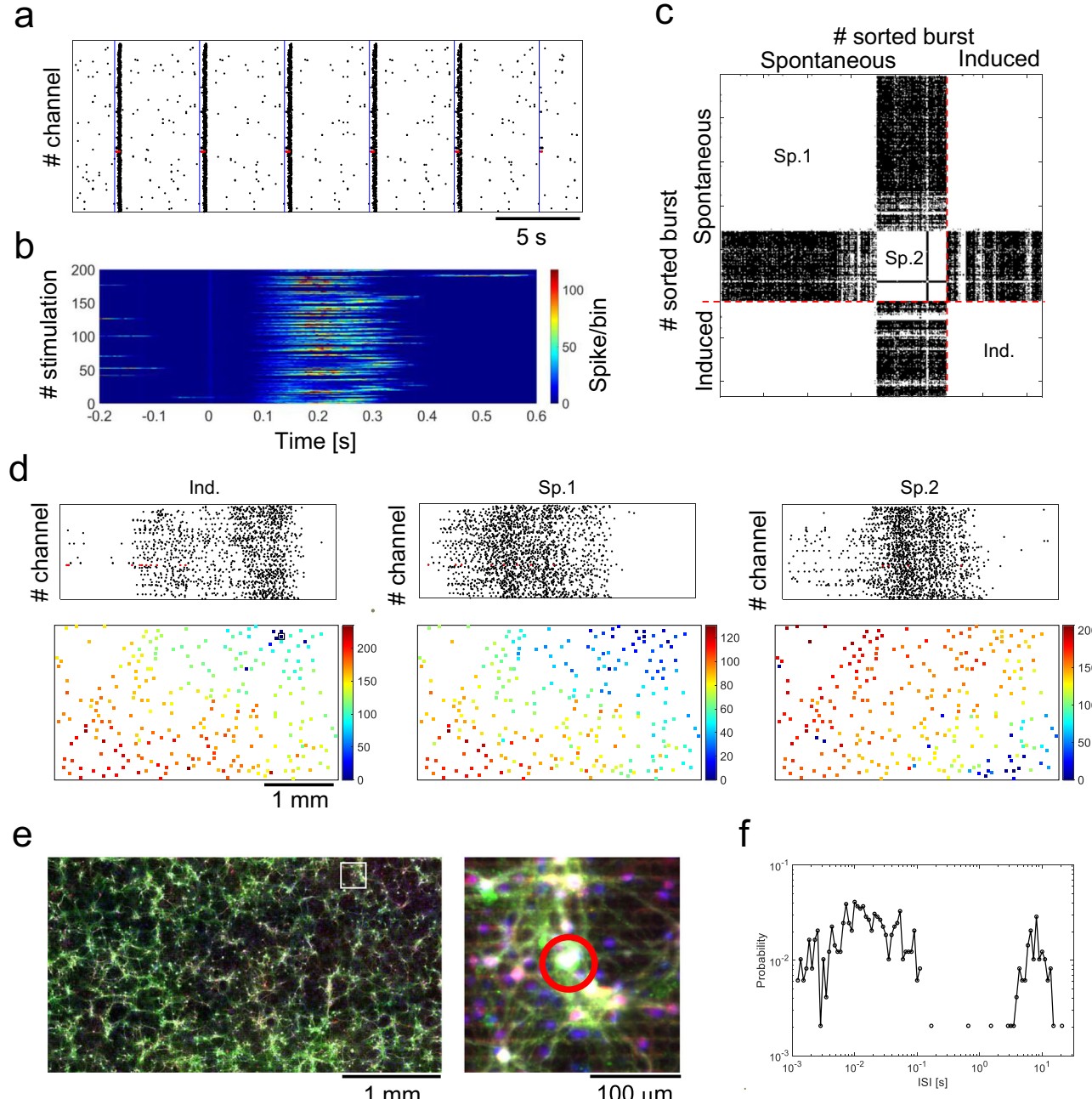

**Fig. 6 | Network bursts initiated by the leader neuron. a** Raster plot during stimulation. The blue line indicates when the leader neuron was stimulated. **b** Firing pattern when the leader neuron was stimulated. Time 0 indicates the stimulation onset. **c** Similarity matrix of burst propagation pattern. Spontaneous bursts were separated into two classes. The propagation pattern of stimulation-induced burst (Ind.) was similar to that of spontaneous class 1 (Sp.1). White color indicates significantly similar burst patterns. **d** Propagation pattern of induced and spontaneous network bursts. Red dots in the spike raster plots indicate the spikes of the leader neuron (top). Latency map of a network burst where the electrode that recorded the activity of the leader neuron in stimulation-induced burst is indicated with a black square (lower). **e** Fluorescence images of the neuronal network. Right panel shows the magnified image (white square area in left panel). The red circle in magnified image indicates the leader neuron. MAP2, gray; GABA, red; GFP-expressing neurons, green; DAPI, blue. Immunofluorescence staining was repeated using two independent samples with similar results. **f** Double logarithmic graph of the leader neuron's inter-spike intervals. The red points in (**a**) and (**d**) indicate the leader neuron activity.

recorded reproducible responses from several dozen neurons stimulated simultaneously using the HD-MEA chip. These data were obtained by configuring a stimulus pattern, initiating electrical recording, and allowing the system to operate autonomously. Consequently, our method is more efficient compared to other techniques such as the patch-clamp method, which demands more manual intervention from the experimenter and can only record a small number of neurons simultaneously. Therefore, the proposed method

is useful for evaluating the response characteristics of multitudinous neurons.

Third, our method offers flexible, targeted stimulation with single-neuron resolution. Here, target neurons were selected with GFP fluorescence and stimulated with $50 \times 50\ \mu m^2$ areas based on the pilot study. The stimulation parameter sufficiently generated reliable direct responses ( > 95%). In addition to the single neuron stimulation, we demonstrated that optical stimulation with a larger area elicited a

broader response (Supplementary Fig. 1) and that the combined stimulation of multiple neurons has a combination effect on response probability (Supplementary Fig. 5). We successfully discovered rare but important neurons with leader and integrator roles owing to flexible and effective stimulation.

Fourth, our method offers a low computational cost for extracting responses from single neurons with a high signal-to-noise ratio. To extract the responses of single neurons from extracellular potential signals, spike source classification (spike sorting) is usually required[62,63]. While spike sorting has been extensively studied and user-friendly software is available, its robustness to changes in spike waveforms remains a challenge. In our method, the responses of single neurons can be extracted without spike sorting through stimulus- or spike-triggered averaging. Therefore, our method is advantageous for extracting single-neuron activity and monitoring changes in response patterns.

The results of this study depend on several experimental conditions. The first is being the volume of the culture medium: HD-MEA usually contains 800 μL of culture medium. The inner diameter of the ring that holds the culture medium is 19 mm, therefore the height of the medium is approximately 2.82 mm. The absorbance of Neurobasal medium for the light at a wavelength of 470 nm, which is used for optical stimulation, is approximately 0.5 at a path length of 1 cm[64]. Therefore, the intensity of the light may change when the volume of the culture medium is different. However, a 10% change in the culture medium changed the light intensity by 3.2%, indicating no significant change. Additionally, a culture medium with low absorbance has been commercially available recent years, which will make optical stimulation experiments more robust with respect to the amount of culture media.

The second is the density of the cultured cells. Higher density results in better network bursts; however, it is difficult to distinguish individual neurons. At low densities, individual cells can be discriminated easily, but the number of surviving neurons is smaller, and network bursts are less likely to occur. In this study, the initial seeding density was fixed at 3000 cells/mm², but the density of cultured neurons around one month in culture could vary depending on the number of times HD-MEA was reused and the viability of the cells at seeding. Therefore, optimizing culture conditions effectively stabilizes the experimental results.

Our study has certain limitations. First, our results were not obtained from intracellular recordings, such as those acquired using the patch-clamp method. The assertion that the activity originates from a single neuron assumes that the timing of stimulus-induced responses in single neurons is consistent and that spike waveforms remain relatively stable. Recent studies have combined HD-MEA with the patch-clamp technique to simultaneously record the electrical activity of multiple single neurons or networks[65,66]. Therefore, combining our method with the patch-clamp technique allows for confident stimulation and recording of single neurons.

Furthermore, the single-neuron network interactions observed in this study are derived from dissociated networks that may not fully mirror the network structure in vivo. Nevertheless, in vitro neuronal networks show scale-free topology[67] and size-distributed neuronal avalanches[68], which is consistent with findings from in vivo and in vitro slices[22,67,69–71]. In addition, in vitro neuronal networks can perform a variety of functions, such as blind source separation and game-playing[72,73]. Thus, our study offers valuable insights into brain networks that are characterized by a combination of bottom-up self-organization and top-down regulation.

In conclusion, we have developed an experimental system by combining optogenetics and HD-MEA electrical recordings, which revealed new aspects of the relationship between single neurons and neuronal networks. We anticipate that this research will serve as a pioneering study within the neuronal network research community, akin to the role of a leader neuron.

## Methods

### Cell culture and gene expression

The rat cortical neurons were cultured on MaxOne HD-MEA chips (MaxWell Biosystems, AG, Zurich, Switzerland). To express Channelrhodopsin 2 (ChR2) and green fluorescent protein (GFP) in neurons, an AAV vector was utilized. All procedures were approved by the University of Tokyo Animal Experiment Committee (KA19-14).

Four days before seeding, the HD-MEA chips underwent hydrophilization and coating following the MaxWell Biosystems protocol. Two days before seeding, 50 μL of 0.1% polyethylenimine (PEI; Sigma-Aldrich Co., Ltd., St. Louis, MO, United States) was applied to the electrodes, and the chips were left overnight at 4 °C. On the seeding day, the PEI solution was removed, and 50 μL of 20 μg/ml laminin (Thermo Fisher Scientific, Waltham, MA, United States) was added to the electrodes. The chips were then incubated at 37 °C for ≥ 1 hour.

Cortical cells from 19-day-old Wistar rat embryos (Charles River Laboratory Japan Inc., Kanagawa, Japan) were dissociated in 0.25% Trypsin (Thermo Fisher Scientific) diluted in Hanks' balanced salts solution (FUJIFILM Wako Pure Chemical Corp., Osaka, Japan) for 17 min at 37 °C. After laminin removal, the cells were seeded on the electrodes at an initial density of 3000 cells/mm². The culture medium consisted of Neurobasal Plus Medium (Thermo Fisher Scientific) supplemented with 500 μM GlutaMax (Thermo Fisher Scientific), 2% B27 Plus Supplement (Thermo Fisher Scientific), and 100 U/ml-100 μg/ml penicillin-streptomycin (Thermo Fisher Scientific). The chips were maintained in an incubator at 37 °C, 5% $CO_2$, and 100% humidity, with half of the culture medium replaced twice weekly.

ChR2 and GFP were expressed using AAV [pAAV-Syn-ChR2 (H134R)-GFP gifted by Edward Boyden, Massachusetts Institute of Technology, MA, United States; Addgene #58880-AAV8]. At 14 days in vitro (DIV), AAV particles were added to the cells at $1.0 \times 10^5$ multiplicity of infection. Two days after adding the AAV, half of the culture medium was replaced.

### Experimental setup

Supplementary Table 1 details the equipment utilized in this study. An overview of the experimental setup is shown in Fig. 1a, b. Optogenetic stimulation was facilitated using a DMD, specifically the Polygon 1000-G Pattern Illuminator system (Mightex Systems, Ontario, Canada). For the polygon's light sources, a blue LED with a center wavelength of 470 nm (LCS-0470-50-22, Mightex Systems) and a green LED with a center wavelength of 560 nm (LCS-0560-68-22, Mightex Systems) were employed.

The HD-MEA recording unit was securely positioned beneath an upright microscope (Olympus Corp., Tokyo, Japan) using a custom jig crafted with a 3D printer (Form3; Formlabs Japan, Tokyo, Japan). HD-MEA chips were situated on the recording unit, and electrical activity data were transmitted via a LAN cable from the recording unit to a recording PC utilizing the MaxOne Hub.

A bespoke black acrylic box was used to shield the experimental system from external light. A thermostat (E5CN, Omron Corp., Kyoto, Japan) was used to maintain the temperature at 37 °C inside the case. To reduce the evaporation of the medium, the HD-MEA chip was covered by custom lids with holes made by the 3D printer. To adjust the $CO_2$ concentration, air from the incubator was supplied above the HD-MEA chips by an air pump (EAP-01, As One Corp., Osaka, Japan). Moreover, to mitigate vibrations, the experimental systems were positioned on a vibration isolator (AVT-0405N; Meiritz Seiki Co., Ltd., Kanagawa, Japan).

## Optical stimulation

Optical stimulation was administered to samples cultured for more than 30 days. The Polygon 1000-G is a DMD, boasts a coverage area of 2200 μm × 3600 μm, and a spatial resolution of 3 μm, facilitated by a 2.5x objective lens (MPLFLN2.5x, Evident Corp., Japan). PolyScan2 (Mightex, CA, USA) was employed for controlling Polygon 1000-G. Due to the bias and variance in stimuli intervals observed in the master mode (Supplementary Fig. 9), the slave mode was used for controlling stimulation timing. In the slave mode, a function generator (AWG1005, As One Corp.) served as the TTL trigger signal source. TTL signals were conveyed from the function generator to the DMD and from the DMD to the recording unit and control module.

The procedure for designing optical stimulation patterns unfolded as follows: First, the position of the recording unit was adjusted using a 2.5x objective lens. The fluorescence images of GFP-expressing neurons were captured using a mirror unit and a microscope camera (WRAYCAM-VEX230M; Wraymer, Osaka, Japan) with a controlling software (MicroStudio, Wraymer). Next, an image of the optical stimulation area was acquired using a green LED that did not excite ChR2 or GFP. A custom MATLAB program facilitated overlaying the fluorescence image and optical stimulation area, x- and y-coordinates of the full irradiated area on the overlayed image were obtained by visually selecting the upper left and lower right. The optical stimulation area was then subdivided into grids of $50 \times 50$ μm$^2$ square with the squares containing neurons expressing ChR2-GFP manually selected. These selected grids were individually subjected to optical stimulation at specific time intervals employing a blue LED. Notably, the optical stimuli were randomly administered, not in the sequence in which the grids were selected.

The LED light intensity was set as a relative percentage of the maximum current (13,000 mA) of the LED, with the intensity fixed at 80%. This intensity at 80% was measured with a power meter (S130VC; Thorlabs Japan, Tokyo, Japan) at 15.4 mW/mm$^2$, which is the same order as in previous studies[31,44].

## Electrical activity recording

Electrical activity recordings were conducted utilizing a MaxOne HD-MEA system. Each HD-MEA chip encompasses 26,400 platinum electrodes ($120 \times 220$ points, 17.5 μm pitch, $9.3 \times 5.45$ μm$^2$ electrode size) within a $3.85 \times 2.10$ mm$^2$ sensing area. The system allows for the simultaneous recording of extracellular potential signals from a subset of 1,024 electrodes at a sampling frequency of 20 kHz with a 10-bit sampling bit rate. Recording was facilitated using the MaxLab Live software provided by MaxWell Biosystems.

Initially, all electrodes were scanned sequentially for 30 s each in activity scan mode while optical stimulation was applied. Next, spike amplitude and time were calculated for each electrode based on the scan result. The recording electrodes were selected as follows.

(1) Neuronal Unit method: This approach aimed to capture the activity of individual neurons while monitoring overall neural network activity on the HD-MEA. A total of 256 units (comprising four electrodes each) exhibiting large average spike amplitudes were selected at a distance of at least 100 μm by MaxLab Live software.

(2) Dense method: Electrodes were densely arranged near the optical stimulation site to delineate local network activity. Spike counts within 15 ms post-stimulus were visualized as a heat map. Subsequently, multiple electrodes exhibiting high spike counts were manually designated as center electrodes. Surrounding these center electrodes, $32 \times 32$ or $22 \times 22$ square grids of electrodes were positioned for recording purposes.

## Immunocytochemistry

The neurons underwent immunostaining to visualize both the neurons and their corresponding electrical activity. Following the electrical activity recording, the medium within the HD-MEA chip was aspirated, and the cells were fixed with 4% paraformaldehyde (PFA; Fujifilm Wako Pure Chemical Corp.) at room temperature for 30 min. After fixation, the PFA was removed, and the HD-MEA chip was rinsed thrice with phosphate-buffered saline (PBS; Thermo Fisher Scientific). The cells were then left at 4 °C overnight. For permeabilization and blocking, a solution comprising 4% Block Ace (Sumitomo Dainippon Pharma, Osaka, Japan) and 0.25% Triton X-100 (Merck, Darmstadt, Germany) diluted in PBS was added to the cells, followed by incubation at room temperature for 3 hours. The primary antibody, diluted in PBS containing 0.4% Block Ace and 0.25% Triton X-100 was added. Then cells were left at 4 °C overnight. Anti-MAP2 antibody (mouse, 1:250; MAB378; Merck, Darmstadt, Germany), anti-NeuN antibody (rabbit, 1:250; ab104225; Abcam, Cambridge, United Kingdom), and anti-B3T antibody (chicken, 1:250; ab41489; Abcam) were used as primary antibodies.

After removing the primary antibody solution, the interior of the HD-MEA chip was triple-washed with PBS. Secondary antibodies diluted in PBS containing 0.4% Block Ace and 0.25% Triton X-100 were applied to the cells, which were then incubated at room temperature for 4 hours. The secondary antibodies used were Alexa Fluor 546 goat anti-rabbit IgG antibody (goat; 1:500; A-11010, Thermo Fisher Scientific), Alexa Fluor 647 goat anti-mouse IgG H&L antibody (goat; 1:500; ab150115, Abcam), and Alexa Fluor 647 goat anti-chicken IgY H&L antibody (goat; 1:500; ab150171, Abcam). Following incubation with the secondary antibody solution, the samples were mounted using Fluoro-KEEPER Antifade Reagent Non-Hardening Type with DAPI (Nacalai Tesque Inc., Kyoto, Japan) and a transparent film (Atto Corp., Tokyo, Japan) after the secondary antibody solution was removed. Confocal microscopy (MAICO MEMS confocal unit; Hamamatsu Photonics, Shizuoka, Japan) and controlling software (HCImage, Hamamatsu Photonics) was employed to observe the cells, with lasers emitting wavelengths of 405 nm, 488 nm, 561 nm, and 638 nm. The confocal image and electrical activity were superimposed using the position of the electrodes observed in the confocal images as a guide.

## Data analysis

Spike data, including spike times, positions of electrodes, and amplitudes, were acquired through real-time spike detection using MaxLab Live. Subsequent analyses were conducted using MATLAB R2021a (MathWorks, Natick, MA, United States). A Butterworth bandpass filter with a cut-off frequency of 300-3500 Hz was applied to filter the recorded extracellular potential signals. This filtering process retained the signals within the spike frequency band while eliminating optical stimulation artifacts. Stimulus time-triggered signals were extracted to evaluate post-stimulus responses. A method from a previous study was used to detect network bursts for network analysis (parameters; $N = 100$, $ISI\_N = 0.2$ s)[74].

The quantification of post-stimulus spike time changes was conducted as follows. First, directly and indirectly responding neurons were selected for reliability. The criteria were a response rate of at least 95% and a distance of less than 70 μm from the stimulation site to the electrode. If more than one electrode met the criteria, one electrode with the highest absolute value of amplitude was selected. Then, if the interval between the end of a network burst and the next network burst was <2 s, they were categorized as a successive network burst.

We defined the BDRC as follows (Supplementary Fig. 10a). The pre-burst response time (pre-BRT) was defined as the average response time to the 10 stimuli preceding the start of the network burst, with the standard deviation of pre-BRT (STD-pre-BRT) calculated accordingly. If another network burst occurred within the 10 stimuli preceding the network burst of interest, only the response times following the end of the other network bursts were considered. The response time to the first stimulus immediately after the end of the network burst was denoted as post-burst response time 1 (post-BRT 1),

with the BDRC representing the absolute difference between post-BRT 1 and pre-BRT. In addition, the BDRC duration was defined as follows (Supplementary Fig. 10b). The second, third, and subsequent response times after the end of the network burst of interest were designated as post-BRT 2, post-BRT 3, etc. If the absolute difference between post-BRT N and pre-BRT exceeded three times the STD-pre-BRT, the BDRC was defined as persisting for N stimuli times after the network burst. If BDRC persisted $N$ times and not $N+1$ times, the BDRC duration was defined as $N$.

Pasquale's method[59] was used to evaluate the similarity of patterns between bursts. For the analysis, we used data measured by arranging the electrodes in a neuronal unit method, with four electrodes as one cluster. First, the electrode with the largest number of firings was selected from each electrode cluster and used in the subsequent analysis. Network bursts were detected as above. In the detected network bursts, the time the earliest firing was detected at each electrode was computed considering the burst start time. The electrodes were arranged in order of earliest firing, and this was used as the propagation pattern of the network burst. For each pair of network bursts, the distance between the propagation patterns was calculated using the edit-distance function in MATLAB. For each length of propagation pattern, we generated 1,000 shuffled datasets and calculated the probability of it being less than or equal to the calculated propagation pattern. When the probability was less than 0.05, the propagation patterns of the pair were considered significantly similar. Finally, network bursts were arranged with the previous method[59].

### Reporting summary

Further information on research design is available in the Nature Portfolio Reporting Summary linked to this article.

## Data availability

All data supporting the findings of this study are available within the article and its supplementary files. Any additional requests for information can be directed to, and will be fulfilled by, the corresponding authors. Source data are provided with this paper. Extracellular potential data and a sample dataset were deposited to Figshare (https://doi.org/10.6084/m9.figshare.26880340).

## Code availability

All custom code used in this manuscript are attached as a zip file.

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

## Acknowledgements

This work was supported by the Japan Society for the Promotion of Science (grant numbers 21K19884, K.S.; 22H03935, Y.J.), the Japan Agency for Medical Research and Development (grant number 21gm6510007h0001, K.S.), and the Murata Science and Education Foundation, K.S.). We especially thank Chie Tamatani, Yuki Miyahara, Takuma Furukawa, and Dai Akita for their helpful discussions.

## Author contributions

T.K., K.S., T.A., and Y.J. designed the project; T.K., K.S., T.N., and T.A. performed the experiments; T.K., K.S., and T.N. analyzed the data; T.K., K.S., and T.N. wrote the manuscript; K.S., K.K., and Y.J. supervised the project. All authors proofread the manuscript and provided comments.

## Competing interests

The authors declare no competing interests.
