## [Transparent Peer Review file · Nature Communications]

Revealing single-neuron and network-activity interaction by combining high-density microelectrode array and optogenetics

Corresponding Author: Dr Kenta Shimba

Version 1:

Reviewer comments:

Reviewer #1

(Remarks to the Author)

Thank you for inviting me to review this manuscript by Kobayashi and colleagues. This study delves into the intricate relationship between single-neuron activity and network synchronization, a fundamental aspect of brain function that has been difficult to reconcile due to inherent limitations imposed by recording apparatuses poised at different scales. Utilizing an innovative experimental setup combining high-density microelectrode array recording with optogenetic stimulation, Kobayashi and colleagues explored cultured networks of rat cortical neurons. This advanced system allowed for simultaneous recording and stimulation at both the single-neuron and network levels, unveiling new dimensions of neuronal behavior. The team discovered a network burst-dependent response change in individual neurons, shedding light on a potential mechanism behind information loss and cognitive impairment during epileptic seizures. Notably, the study also documented the first direct recording of a "leader" neuron initiating a spontaneous network burst, highlighting the pivotal role of hub neurons in driving synchronization. These findings offer profound insights into brain network dynamics, illustrating a complex interplay of self-organization and regulatory processes that underpin brain function.

I found the work to be methodologically-sound, significant and potentially noteworthy for the field, particularly with respect to the advances borne by the development of the apparatus used in the study.

While I enjoyed the work, I'm not sure that I appreciated the authors' comment regarding: "experimental system capable of analyzing at the single-neuron and network scale". If the authors are referring to the capacity to record precise neuron-level activity that can be scaled to the level of the whole system, then this should be stated more explicitly.

I found myself wondering whether individual cellular activity was correlated with summary statistics of the network-level signatures. Did the authors consider quantifying distinctly network-level signatures and reporting them along with the primary responsiveness of the neurons themselves? For instance, measures from graph theory (e.g., modularity) or statistical measures (e.g., PCA) could help to augment the conclusions that you could make at the end of your study.

Have the authors considered making their data open-source following publication? This could help to improve the potential impact of their work for the field.

Reviewer #2

(Remarks to the Author)

In this work, the authors present an experimental setup to investigate cell culture dynamics by combining high density electrical recordings with optical stimulation, reaching the single cell level.

The capabilities of the setup are nicely presented along the paper. However, at the current stage, the data analysis part is really poor and a potential reader cannot understand if the recording system allows to record reliable and stable responses of the network. As such, I believe this work would certainly be of interest for a more specific and technical journal than Nature Communication. However, I also think their setup could be used to sustain in a more rigorous fashion their results. I

therefore invite the authors to carefully revise the entire manuscript in order to present their results in a more organic way, according to the review reported below. In addition, the authors should clarify the extent of some terms used along the manuscript. For instance, the term synchronous is associated to the emergence of a network burst, while the latter is a propagating event and not a synchronous event. Since the authors can record spiking event, the time scale for synchronous events is on the order of the duration of a spike (aka 1 ms) and not of a network burst (>100 ms).

Major comments

Question 1

page 10, lines 3-5: From description and Figure 2 it seems that single light pulses of 5ms duration induce robust activation of the network lasting for the entire relaxation time window (1s). However in the literature evoked activities following an electrical stimulation have been reported to last for less than 400 ms (see Nieus et al. Sci Report 2018). Please report a zoom of the 1s-time window to clarify this point and explain the reason of the observed difference.

Question 2

page 10, line 6: The 1 Hz stimulation frequency is typically used to induce LTD. Did the authors checked if any plasticity was induced? Please report analysis.

Question 3

page 10, line 10: "Synaptically evoked indirect responses have also been observed", this information is too anecdotal. The authors should report how many direct and indirect responses have been found as this information is important also to understand what region/neurons are ready to respond to stimulation and elicit network bursts.

Question 4

page 11, lines 2-9: The author present a result (again anecdotal!), that it is not really unexpected. Please: 1) provide more examples 2) remove the sentence: "This result suggests that combined stimulation will unravel new information processing activity in living neuronal networks."

Question 5

page 11, lines 16-18, Page 12 Lines 1-7. With respect to the recorded data, what is the fraction of integrator neurons?

Question 6

With respect to the BDRC analysis. At page 12, lines 1-2: "The BDRC trend was reproducible in the same neurons" Please give a quantification. Again, at page 12, lines 17-18: "the post-stimulus spike times changed discontinuously and then recovered continuously" the figures are unclear (each dot indicates what?) and the description does not add much information (Legend Fig.5 "Post-stimulus raster plots at electrodes with BDRC" with respect to a single electrodes, a selection, or which one?). Maybe another quantification (variance?) along time would help to clarify it.

Question 7

The concept of leader neurons has been explored by others. For instance, in Pasquale et al. 2017 (this paper should be cited!) the authors demonstrate, corroborated by spike sorting analysis, that the leader neurons are responsible of the network bursts during the spontaneous activity and are also involved in the evoked event.

Again, the authors present an anecdotal case of a leader neuron. Again it is not clear if this occurred in just a single experiment. The lack of quantification, as confirmed by the sentence "We successfully discovered rare but important neurons with "leader" and "integrator" roles owing to flexible and effective stimulation." lowers significantly the value of this work.

Moreover the leader neuron was activated by optical stimulation but effective leader neurons have also been found during recordings of pure spontaneous activity. I invite the authors to perform more rigorous analysis, that is coherent with the current state of the art (see Pasquale et al. 2017). Moreover, the authors should also extend the discussion to this respect as the leader neurons are likely embedded in a portion of the network with some special properties. For instance, modeling works (Lonardoni et al. 2017) have shown that network burst presumably originate from areas with a high connectivity and these regions are susceptible to sub-threshold stimulation. To this regard I invite the authors to perform additional experiments to verify the hypothesis that the stimulation of the neurons surrounding the leader one can also generate network bursts.

The red x of Figure 6a,d,e are not explained. I guess they refer to the time of stimulation of a given channel/neuron. If this is correct the sentence: "Importantly, the leader neuron-initiated network bursts during spontaneous activity (see Fig. 6d)" (page 14, lines 6-7) is not correct as the activity is actually induced by a stimulus.

With respect to the following (page 14): "Through immunostaining after recording, we successfully identified leader neurons (refer to Fig. 6c), suggesting that a detailed analysis of protein or gene expression patterns can also be analyzed in detail.", I do not think the sentence pertains to the results, maybe in the discussion.

Minor comments

1. Page 12 Line 18 Panel of Figure 5b vertical bar is mV not ms
2. Why is the description of "Supplementary Fig. 3" reported in yellow?
3. At page 16 line 8. The sentence: "This method has significantly contributed to various electrophysiological studies." does not make sense without references.
4. Page 20, line 10: "In recent studies, HD-MEA has combined with the patch-clamp" please correct

References

Pasquale, V., Martinoia, S., and Chiappalone, M. (2017). Stimulation triggers endogenous activity patterns in cultured cortical networks. *Scientific Reports* 7. doi: 10.1038/s41598-017-08369-0

Lonardoni, D., Amin, H., Di Marco, S., Maccione, A., Berdondini, L., and Nieuws, T. (2017). Recurrently connected and localized neuronal communities initiate coordinated spontaneous activity in neuronal networks. *PLOS Computational Biology* 13, e1005672. doi: 10.1371/journal.pcbi.1005672

Reviewer #3

(Remarks to the Author)

Kobayashi et al. present an experimental framework that combines optical stimulation and HD-MEA recordings. The authors discuss the interesting problem of understanding to what extent single-neuron activity is related to network one, which is of possible interest for the journal reader. The authors apply the framework to cortical neurons cultured in monolayers.

The results are promising, but not as convincing. The cited literature in the introduction and discussion could be more elaborated and complete. The Matlab code seems to me to be not available. The description and illustration of the experimental setup in Figure 1 could be improved. The steps performed should be more clearly stated.

Although the manuscript has some potential, it falls short at several points. e.g. it is not discussed how cell density and the experimental setup (e.g., volume of medium) might influence the performance of what they are proposing. This is the major limitation for the performance assessment. To me, the manuscript appears to be premature and at the current stage not suitable for Nature Communications.

Version 2:

Reviewer comments:

Reviewer #1

(Remarks to the Author)

The authors have adequately addressed my concerns.

Reviewer #2

(Remarks to the Author)

The authors performed new experiments and added a significant amount of analysis to the manuscript. I am happy that my review has been addressed carefully. Although I believe the data analysis part could have been further extended, the updated manuscript has improved considerably. In line with the idea of 'open science' data I have also much appreciated the fact the authors are sharing the code and data to reproduce all figures of the manuscript. I think the manuscript can now be accepted. Good work!

Reviewer #3

(Remarks to the Author)

REVIEWER COMMENTS

Reviewer #1 (Remarks to the Author):

Thank you for inviting me to review this manuscript by Kobayashi and colleagues. This study delves into the intricate relationship between single-neuron activity and network synchronization, a fundamental aspect of brain function that has been difficult to reconcile due to inherent limitations imposed by recording apparatuses poised at different scales. Utilizing an innovative experimental setup combining high-density microelectrode array recording with optogenetic stimulation, Kobayashi and colleagues explored cultured networks of rat cortical neurons. This advanced system allowed for simultaneous recording and stimulation at both the single-neuron and network levels, unveiling new dimensions of neuronal behavior. The team discovered a network burst-dependent response change in individual neurons, shedding light on a potential mechanism behind information loss and cognitive impairment during epileptic seizures. Notably, the study also documented the first direct recording of a "leader" neuron initiating a spontaneous network burst, highlighting the pivotal role of hub neurons in driving synchronization. These findings offer profound insights into brain network dynamics, illustrating a complex interplay of self-organization and regulatory processes that underpin brain function.

I found the work to be methodologically-sound, significant and potentially noteworthy for the field, particularly with respect to the advances borne by the development of the apparatus used in the study.

Response: We appreciate your assessment of our paper. Your valuable comments have further enhanced the completeness of this manuscript. Below are our responses to your comments.

While I enjoyed the work, I'm not sure that I appreciated the authors' comment regarding: "experimental system capable of analyzing at the single-neuron and network scale". If the authors are referring to the capacity to record precise neuron-level activity that can be scaled to the level of the whole system, then this should be stated more explicitly.

Response: As noted by Reviewer 1, the subject of this paper, single neuron and network scale analysis, can record activity at the precise neuron level and extend it to the system-wide level. The following corrections have been made to clarify the meaning of the text.

Abstract (P.2, L.7)

we established a new experimental setup enabling simultaneous recording and stimulation at a precise single-neuron level that can be scaled to the level of the whole network.

Introduction (P.4, L.11)

To clarify these interactions, an experimental system that can analyze whole neuronal networks at the single-neuron resolution, is essential.

Introduction (P.5, L. 15)

Recently, high-density microelectrode arrays (HD-MEAs) based on CMOS technology have provided simultaneous long-term recordings at single-neuron resolution in neuronal networks

Discussion (P.18, L.1)

Optogenetic stimulation with single-neuron resolution induced activity in individual neurons, while HD-MEA simultaneously recorded activity at single-neuron resolution in the whole network.

I found myself wondering whether individual cellular activity was correlated with summary statistics of the network-level signatures. Did the authors consider quantifying distinctly network-level signatures and reporting them along with the primary responsiveness of the neurons themselves? For instance, measures from graph theory (e.g., modularity) or statistical measures (e.g., PCA) could help to augment the conclusions that you could make at the end of your study.

Response: As you suggested, we created a map of functional connections based on graph theory and classified the node neurons into sender and receiver. Consequently, we found that the leader neuron belongs to the sender group and the surrounding cells are also senders of information (Supplementary Fig. 7). Additionally, we calculated the pair-wise similarity between every combination of network bursts, a statistical measure of network bursts, and compared spontaneous network burst and stimulation-induced burst. As shown in the modified version of Figure 6, the spatiotemporal pattern of bursts generated when the leader neuron was stimulated matched one of the spatiotemporal patterns of spontaneously generated bursts. This result is consistent with previous studies (Pasquale et al., 2017) and suggests that bursts generated by stimulation occur spontaneously in networks formed by self-organization. These strongly reinforce the conclusions of this paper. We have added Supplementary Fig.7 and revised the Fig.6 and Results section.

Results (P.16, L. 11)

To characterize the properties of the leader neuron, we examined functional connections among the network and its inter-spike interval (ISI). The leader neuron was classified as the information sender, and network activity was generated from a cluster of sender neurons including the leader neuron (Supplementary Fig. 7).

Supplementary Fig.7

Supplementary Fig. 7.

Map of Functional connectivity. Pair-wise functional connectivity was calculated for all pairs of electrodes. Information sender and receiver was defined with the direction of functional connection. Mean ratio of sender and receiver roles was indicated with color; warm color means sender, cold color means receiver. The black arrow indicates the position of the leader neuron, which works as an information sender in the network.

Have the authors considered making their data open-source following publication? This could help to improve the potential impact of their work for the field.

Response: We have attached the MATLAB code used in the analysis along with the sample data as a zip file. We thank the Reviewer for this suggestion to make the data and code open, which increases the value of our research.

Reviewer #2 (Remarks to the Author):

In this work, the authors present an experimental setup to investigate cell culture dynamics by combining high density electrical recordings with optical stimulation, reaching the single cell level.

The capabilities of the setup are nicely presented along the paper. However, at the current stage, the data analysis part is really poor and a potential reader cannot understand if the recording system allows to record reliable and stable responses of the network. As such, I believe this work would certainly be of interest for a more specific and technical journal than *Nature Communication*. However, I also think their setup could be used to sustain in a more rigorous fashion their results. I therefore invite the authors to carefully revise the entire manuscript in order to present their results in a more organic way, according to the review reported below. In addition, the authors should clarify the extent of some terms used along the manuscript. For instance, the term synchronous is associated to the emergence of a network burst, while the latter is a propagating event and not a synchronous event. Since the authors can record spiking event, the time scale for synchronous events is on the order of the duration of a spike (aka 1 ms) and not of a network burst (>100 ms).

Response: We appreciate your valuable comments and appropriate understanding of the paper. We have improved the quality of the paper by adding quantitative data and defining terms appropriately according to your comments. The content of the paper is of interest to a wide range of readers, including those interested in brain self-organization and the relationship between single neurons and networks, as commented by Reviewers 1 and 3. In line with Reviewer 2's comments, we believe that the addition of quantitative data and results further strengthens the conclusions and makes the content suitable for *Nature Communications*.

Regarding word synchronization, the word synchronous activity is widely used even when spikes are detected. Conversely, as the duration of a spike and the time of a network burst are on different scales, we must choose the word more carefully in this study, focusing on propagation within network activity, as pointed out by Reviewer 2. Therefore, we have revised the manuscript to exclude the word "synchronization" and used the "network activity" or "network bursting" as suggested.

Major comments

Question 1

page 10, lines 3-5: From description and Figure 2 it seems that single light pulses of 5ms duration induce robust activation of the network lasting for the entire relaxation time window (1s). However in the literature evoked activities following an electrical stimulation have been reported to last for less than 400 ms (see Nieus et al. Sci Report 2018). Please report a zoom of the 1s-time window to clarify this point and explain the reason of the observed difference.

Response: In Figure 2, no network activity occurred because the leader neuron was not stimulated. Owing the higher spatial specificity of optical stimulation compared to electrical stimulation, there are cases where stimulation near a leader neuron does not induce activity of the leader neuron. In the data shown in Figure 2, 122 neurons were stimulated at intervals of 1/122 s, with a stimulus frequency of 1 Hz for each neuron. Therefore, the raster plot in Figure 2 shows a similar pattern every second. As Reviewer 2 suggested, we have added a magnified version of Fig. 2b as Fig. 2c. A two-second time window was chosen to show the multi-cycle response pattern. Additionally, stimulus pattern and experimental procedure have been added to Fig. 1 to make the stimulus conditions clearer. We have also modified the manuscript for better understanding.

Results (P.9, L.5)

To determine the stimulation areas, fluorescence images of GFP was taken (Fig. 1d, step 1). The stimulation area were divided into grids and superimposed with the GFP fluorescence image (Fig. 1d, step 2). Grids containing ChR2-GFP-expressing neurons were manually selected (Fig. 1d, step 3). Optical stimulus was delivered to each selected location sequentially (Fig. 1e).

Result (P.10, L.6)

Additionally, minimal jitter responses of 79 neurons out of the 122 stimulated neurons were simultaneously recorded from the network (see Fig. 2b) with high reproducibility (Fig.2c).

Figure 1

Figure 2c

Figure.2 caption

(c) Magnified raster plot. Representative stimulation times are indicated with green, blue and red vertical lines. Because the stimulation frequency for each neuron was 1 s optical stimulus was applied to the same location at a 1-s interval.

Question 2

page 10, line 6: The 1 Hz stimulation frequency is typically used to induce LTD. Did the authors check if any plasticity was induced? Please report analysis.

Response: We have checked the stability of latency and probability of indirect responses. When LTD is induced, characterizing neurons that respond via synaptic connections becomes difficult because synaptic transmission efficacy decreases. Hence, it is important to confirm that LTD has not occurred. We have evaluated the response latency and response probability of indirectly responding neurons in four samples stimulated at 1 Hz for 50 minutes.

Response latency decreased and response probability increased for 5-10 min after the stimulus was initiated. The same trend was observed afterwards, but the change was less than 5%. These results indicate that the stimulation may have caused a kind of slowly changing plasticity, although no LTD was induced. The possible reasons for the plasticity include adaptation to repeated stimuli and the termination of the increase in activity level due to the disturbance when the samples are moved from the incubator to the recording system. However, because response probability was maintained at a high level (>0.95), change in response latency was small ($<5\%$), and the slow change compared to the change associated with the network burst, this plasticity does not affect the following results of this study. An explanation for the above phenomenon has been added to the Results sections.

Results (P.11, L. 7)

Previous study reported that electrical stimulation with the frequency of 1 Hz induced long-term depression (LTD). When LTD is induced, characterizing neurons that respond via synaptic connections becomes difficult because synaptic transmission efficacy decreases. Thus, we evaluated the response latency and probability (Supplementary Fig. 4). Response latency decreased and response probability increased for 5-10 min after the stimulus was initiated. The same trend was observed afterwards, but the change was less than 5%. These results indicate that the stimulation may have caused a kind of plasticity, such as synaptic plasticity or facilitation, although no LTD was induced. However, because response probability was maintained at a high level (>0.95), change in response latency was small ($<5\%$), and the slow change compared to the change associated with the network burst, this plasticity does not affect the following results of this study.

Supplementary Figure 4

Supplementary Fig. 4. Change in response latency and probability.

(a) Time dependent change in response latency and probability for a representative sample. The latency decreased during the recording period. (b) Change in normalized response latency. Latency was averaged every 300 s and normalized the overall average value at each neuron. Horizontal dashed lines show the average $\pm 5\%$. Mean \pm standard deviation. (c) Change in response probability. Response latency decreased and response probability increased for 5-10 min after the stimulus was initiated.

Question 3

page 10, line 10: “Synaptically evoked indirect responses have also been observed”, this information is too anecdotal. The authors should report how many direct and indirect responses have been found as this information is important also to understand what region/neurons are ready to respond to stimulation and elicit network bursts.

Response: In the present experiment, optic stimulation was applied to two independent samples with the electrode arrangement of the dense method, and direct and indirect responding cells were quantified. When stimuli were applied to 66 locations, 66 cells responded directly and 125 cells responded indirectly. To show which areas had cells ready to respond to the stimuli, we have included a supplementary figure showing the arrangement of direct and indirect responses and stimulation locations. Based on the above, the results and supplement files have been added.

Result (P.11, L.1)

In two independent samples, we stimulated 66 locations, and detected 66 directly responding neurons and 125 indirectly responding neurons. The responding neurons and optical stimulation locations are shown in Supplementary Figure 3.

Question 4

page 11, lines 2-9: The author present a result (again anecdotal!), that it is not really unexpected. Please: 1) provide more examples 2) remove the sentence: “This result suggests that combined stimulation will unravel new information processing activity in living neuronal networks.”

Response: An important aspect of this result is that there were electrodes for which the probability of a response decreased with combined stimulation. Quantitatively, from two different recording positions, 8.1% and 5.8% of the electrodes had an increase in the number of responses above 20%, and 1.5% and 1.2% had a decrease. These results have been added with another example, and the last sentence in the paragraph have been deleted as Reviewer 2 has suggested.

Results (P.12, L.7)

From the difference between aggregate response and response to combined stimulation,

we found both electrodes with increased and decreased activity with the combined stimulation (>20% increase, 8.1% and 5.8%; <20% decrease, 1.5% and 1.2%; two different electrodes settings).

Supplementary Fig.5

Supplementary Fig. 5. Response probability of single-site and combined stimulation.

The single-site stimulations at sites A and B and combined stimulation at both sites A and B were delivered sequentially. (a) Firing probability to stimulus with four different conditions. The figures show the results for, from left to right, response to stimulus A, stimulus B, aggregate response obtained by summing the responses of stimulus A and B, and combined stimulus A+B. (b) Difference between aggregate response and combined stimulation. Red color in (b) shows the response probability was higher with combined stimulation than aggregate response of stimulus A and B. Note that a few electrodes show a decrease in firing probability with the combined stimulus.

Question 5

page 11, lines 16-18, Page 12 Lines 1-7. With respect to the recorded data, what is the fraction of integrator neurons?

Response: Of the indirectly responding neurons, 4% were neurons integrating two stimuli, and 0.8% were neurons responding to three stimuli (n = 130 neurons). These results have been added to the Results section.

Results (P.13, L.7)

Of the indirectly responding neurons, 4% were neurons integrating two stimuli, and 0.8% were neurons responding to three stimuli (n = 130 neurons).

Question 6

With respect to the BDRC analysis. At page 12, lines 1-2: “The BDRC trend was reproducible in the same neurons” Please give a quantification. Again, at page 12, lines 17-18: “the post-stimulus spike times changed discontinuously and then recovered continuously” the figures are unclear (each dot indicates what?) and the description does not add much information (Legend Fig.5 “Post-stimulus raster plots at electrodes with BDRC” with respect to a single electrodes, a selection, or which one?). Maybe another quantification (variance?) along time would help to clarify it.

Response: Analysis and explanation of BDRC have been added. First, BDRC is a phenomenon in which the latency from stimulus to response changes after a network burst. In this paper, BDRC was calculated as the time difference between the response latency after a network burst and the average of 10 latencies before the burst. To qualitatively demonstrate this phenomenon, a graph with the measurement time on the horizontal axis and the latency on the vertical axis is suitable, and readers can compare the change in latency with the time of network burst occurrence. Figures 5a and 5d show graphs created by the above procedure for a representative electrode at which BDRC was observed, with each dot indicating the latency to a single stimulus. The latency change and recovery can be seen after the red lines indicating the network bursts; Fig. 5b has been modified to clearly show the change in BDRC.

To evaluate the reproducibility of BDRC, neurons exhibiting BDRC were sorted in order of highest mean BDRC, and all BDRC values were plotted. The top 36%

(26 of 73 neurons) showed a prolonged mean latency of >1 ms, with 92.7% of BDRC events being prolonged latencies. On the other hand, only 5.5% (4 of 73 neurons) showed a shortening of the mean latency, which tended to have a large standard deviation. These results suggest that network bursts normally increase response latency. Based on these results, the Results section was revised.

In addition, we calculated the coefficient of variation (CV) in latency every 10 seconds for each electrode to show the variance over time. Figure 5e top panel shows time-dependent change in mean CV and the bottom panel shows the heatmap of each neuron with time on the horizontal axis and electrode number on the vertical axis. As a result, the CV temporarily increased at the timing of the bursts indicated by the red vertical lines, making it easier to see the changes in the entire electrodes. Based on the above modifications, Fig. 5 and the caption were revised, and an explanation was added to the Results section.

Results (P.14, L.5)

Subsequently, the burst-dependent changes were quantified (see Methods). First, coefficient of variation (CV) of response latency was calculated for evaluating the variance over time. Network burst globally increased CV of the response latency in the network (Fig. 5e). Then, to evaluate the reproducibility of burst-dependent changes, burst-dependent response change (BDRC) was calculated as the time difference between the response latency after a network burst and the average of 10 latencies before the burst. Neurons exhibiting BDRC were sorted in order of highest mean BDRC, and all BDRC values were plotted (Fig. 5f). The top 36% of neurons (26 out of 73 neurons) showed a prolonged mean latency of >1 ms, with 92.7% of BDRC events being prolonged latencies. On the other hand, only 5.5% of neurons (4 out of 73 neurons) showed a decrease in BDRC, which tended to have a large standard deviation. These results suggest that network bursts generally prolonged response latency and increase the variance.

Revised Figure 5 and figure caption

Fig. 5. Burst-dependent response change.

(a) Burst-dependent change in response latency to optical stimuli at a representative electrode. The red vertical lines indicate the time of network bursts. Top, prolonged latency; bottom, shortened latency. (b) Enlargements of (a). (c) extracellular potential waveforms triggered at stimulus before and after network bursts in (b). Pre-Burst indicates response to stimulus immediately before the network burst. Post-Bursts 1, 2, 3, and 4 indicate response to 1st, 2nd, 3rd, and 4th responses immediately after the network burst. The horizontal blue bar represents the stimulus duration (5 ms). (d) An example of the long-lasting change in response latency after a successive network burst. (e) Burst dependent change in variance of response latency. Coefficient of variation (CV) was computed at each neuron. Top, mean CV over all neurons; bottom, heatmap for each neuron. Note that network burst globally increased CV

of response latency in the network. (f) Sorted burst-dependent response change (BDRC) for each neuron. BDRC was calculated as the time difference between the response latency after a network burst and the average of 10 latencies before the burst. Each dot indicates each BDRC value. Mean \pm standard deviation. (g) Comparison of BDRC between one and successive bursts. (h) Comparison of BDRC durations between one and successive bursts.

Question 7

The concept of leader neurons has been explored by others. For instance, in Pasquale et al. 2017 (this paper should be cited!) the authors demonstrate, corroborated by spike sorting analysis, that the leader neurons are responsible of the network bursts during the spontaneous activity and are also involved in the evoked event.

Response: Pasquale et al. found the “leader site” because they used low-density MEA. However, in this study, we found the “leader neuron” because of the sufficient spatial resolution of HD-MEA. We have cited the paper and added a comparison with previous studies in the discussion section.

Discussion (P.20, L.5)

Pasquale et al. recorded network activity with conventional MEA with 64 electrodes and showed that “leader electrodes” were rapidly recruited within both spontaneous and electrically induced bursts⁵⁸. However, it remains unclear whether a single neuron can spontaneously and intrinsically initiate network activity across the entire network.

Again, the authors present an anecdotal case of a leader neuron. Again it is not clear if this occurred in just a single experiment. The lack of quantification, as confirmed by the sentence “We successfully discovered rare but important neurons with “leader” and “integrator” roles owing to flexible and effective stimulation.” lowers significantly the value of this work.

Response: To evaluate the presence of burst leaders, we applied 100 stimuli with a frequency of 0.2 Hz to 8 different samples (20-40 locations to each sample) prepared at three different times. The results showed that 4 out of the 8 samples (4 locations) elicited bursts with a probability of 10% or greater (10, 18, 60, 98%). To clearly present the quantitative data, the number of stimulation locations which induced a network burst,

the probability of eliciting a network burst, and the number of stimulus locations have been added to the Results section.

Results (P.15, L.10)

Finally, we examined how specific neurons with hub roles initiate a network **burst**. As shown in Fig. 6a, **Fig. 6b**, and Supplementary Video 3, 1.3% of **neurons** (4 locations out of 290 stimulation locations from eight independent samples) initiated network bursts when stimulated with a pulse of 0.2 Hz, with a network burst initiation probability of 10%, 18%, 60%, and 98%.

Moreover the leader neuron was activated by optical stimulation but effective leader neurons have also been found during recordings of pure spontaneous activity. I invite the authors to perform more rigorous analysis, that is coherent with the current state of the art (see Pasquale et al. 2017). Moreover, the authors should also extend the discussion to this respect as the leader neurons are likely embedded in a portion of the network with some special properties. For instance, modeling works (Lonardoni et al. 2017) have shown that network burst presumably originate from areas with a high connectivity and these regions are susceptible to sub-threshold stimulation. To this regard I invite the authors to perform additional experiments to verify the hypothesis that the stimulation of the neurons surrounding the leader one can also generate network bursts.

Response: We have performed additional analyses and experiments following your comments.

We have added an analysis using Pasquale's method to evaluate the relationship between spontaneous activity and network bursts induced by optic stimuli. The results confirmed that a single pattern of activity is produced by the stimulus and that the stimulus-induced activity is part of the repertoire of spontaneous activity. We have modified Figure 6 and the manuscript to include these results.

Furthermore, we conducted the proposed experiment. A summary of the experiment is as follows, and also shown in Supplementary Figure 8: stimuli of 50 μm squares were applied to five adjacent areas (target plus one square shifted up, down, left, and right), and the responses were measured. With this experiment, we detected network activity that was elicited with a probability of 10% or greater when two of the four samples were stimulated at a total of four locations. Network activity was elicited even

when stimulating squares other than the one that elicited the burst with the highest probability, and analysis using Pasquale's method showed that a similar pattern of network activity occurred when shifting by one square. However, the probability of occurrence decreased, and the time lag between the stimulus and the onset increased with the shift. In addition, the electrode order of firing tended to be different for the electrodes near the stimulus. These results are consistent with a previous study. They show that processing by the surrounding local network results in signals being transmitted to a leader neuron near the stimulus location that generate network activity with the highest probability and shortest time delay.

The sensitivity to subthreshold stimulation was set up assuming the simultaneous stimulation of about 40 neurons in the simulation conditions of the previous study (Lonardoni et al. 2017). When multiple cells are stimulated simultaneously, the stimulation effect will be amplified by synaptic transmission between cells. However, it is unlikely that local amplification by synaptic transmission occurs in the stimulation of a single cell in this study. Therefore, the effects of subthreshold stimulation should be evaluated by stimulating numerous cells. We did not perform the wide-area stimulation because it is outside of the scope of this study, which targets single-cell stimulation. However, the additional experiments described above suggest that the spikes were amplified by the local circuit, which is consistent with the results of the modeling study. Subthreshold stimulation can be realized in the future by optimizing the stimulus range and intensity.

These contents have been added to the Results section. This experiment clarified the relationship between leader neurons and local networks in this paper and significantly increased the value of the paper. We thank Reviewer 2 for the valuable advice.

Results (P. 17, L.3)

To further characterize the burst generating local circuits, optic stimuli were applied to the target locations and 50 μm shifted locations (Supplementary Fig. 8a), for a total of five locations. Results showed that bursts were sometimes induced by stimulation of nearby locations (Supplementary Fig. 8b). Pattern analysis showed that bursts induced by stimuli in close proximity showed a similar propagation pattern (Supplementary Fig. 8cd). On the other hand, the electrode order of firing tended to be different for the electrodes near the stimulus (Supplementary Fig. 8c). Latency to network bursts was calculated and found to be significantly different for each stimulus location (Supplementary Fig. 8ef). These results suggest that local neural circuits are involved in the generation of network bursts.

Figure 6

Fig. 6. Network bursts initiated by the leader neuron.

(a) Raster plot during stimulation. The blue line indicates when the leader neuron was stimulated. (b) Firing pattern when the leader neuron was stimulated. Time 0 indicates the stimulation onset. (c) Similarity matrix of burst propagation pattern. Spontaneous bursts were separated into two classes. The propagation pattern of stimulation-induced burst (Ind.) was similar to that of spontaneous class 1 (Sp.1). White color indicates significantly similar burst patterns. (d) Propagation pattern of induced and spontaneous network bursts. Red dots in the spike raster plots indicate the spikes of the leader neuron (top). Latency map of a network burst where the electrode that recorded the activity of the leader neuron in stimulation-induced burst is indicated with a black square (lower). (e) Fluorescence images of the neuronal network. Right panel shows the magnified image

(white square area in left panel). The red circle in magnified image indicates the leader neuron. MAP2, gray; GABA, red; GFP-expressing neurons, green; DAPI, blue. (f) Double logarithmic graph of the leader neuron's inter-spike intervals. The red points in (a) and (d) indicate the leader neuron activity.

Supplementary Figure

Supplementary Fig. 6. Relationship between stimulation location and leader neurons. (a) Locations of recording electrodes and stimulation. Optic stimulation was applied to five adjacent areas shown by the red square (target plus one square shifted up, down, left, and right). Electrodes are shown with yellow squares. (b) Firing rate after optic stimulation. Network bursts were induced by optic stimulation to Up location. Yellow vertical lines show the stimulation time. (c) Burst propagation patterns induced by optic stimulation to Up and Right locations. The overall pattern of propagation was similar (right panels), but the response time was different for the electrodes closer to the stimulus locations (left panels). Red squares indicate stimulation locations. (d) Similarity matrix of burst propagation pattern. Induced bursts from Up location (Ind. 1) and Right location (Ind.2) are significantly similar to Spontaneous burst 6 (Sp.6). (e) Calculation method for latency of network burst. (f) Comparison among burst latency. Latency of network bursts induced by optic stimulation to Up location was significantly shorter than that to Right location. ***, $p < 0.001$; Mann–Whitney U test; $n = 49$ bursts from Up location, $n = 11$ bursts from Right location.

The red x of Figure 6a,d,e are not explained. I guess they refer to the time of stimulation of a given channel/neuron. If this is correct the sentence: “Importantly, the leader neuron-initiated network bursts during spontaneous activity (see Fig. 6d)” (page 14, lines 6-7) is not correct as the activity is actually induced by a stimulus.

Response: The red x indicates the activity time of the leader neuron, not the stimulus time. As Figures 6e and f show the results of spontaneous activity, Figure 6e shows that network bursts occur because of spontaneous activity of the leader neuron. The explanation for the red x has been added to the caption of Figure 6.

Figure 6 legend

The red points in (a) and (d) indicate the leader neuron activity.

With respect to the following (page 14): “Through immunostaining after recording, we successfully identified leader neurons (refer to Fig. 6c), suggesting that a detailed analysis of protein or gene expression patterns can also be analyzed in detail.”, I do not think the sentence pertains to the results, maybe in the discussion.

Response: We have moved the latter part of the sentence to the Discussion section.

Discussion (P.20, L.14)

Additionally, because the leader neuron could be identified with immunostaining after recording, detailed analysis of protein or gene expression patterns can also be performed for further characterizing the leader neuron.

Minor comments

1. Page 12 Line 18 Panel of Figure 5b vertical bar is mV not ms

Response: The index BDRC introduced in this paper indicates the change in latency of the response to the light stimulus after the burst has occurred. Therefore, the vertical scale bar in Fig. 5b is correct in ms. On the other hand, because the scale bar in Fig. 5c was not attached, a vertical scale bar in unit μV have been added.

2. Why is the description of “Supplementary Fig. 3” reported in yellow?

Response: This manuscript was resubmitted after rejection and was highlighted in yellow to clarify the revisions at the time of resubmission. In the revised version, the highlight on “Supplementary Figure 3” has been removed, and only the revisions reflecting this peer review have been highlighted.

3. At page 16 line 8. The sentence: “This method has significantly contributed to various electrophysiological studies.” does not make sense without references.

Response: We have added a reference and revised the manuscript.

Discussion (P.18, L.15)

This method has significantly contributed to various electrophysiological studies⁵⁰, including this study.

4. Page 20, line 10: “In recent studies, HD-MEA has combined with the patch-clamp” please correct

Response: We have modified the sentence as follows.

Discussion (P.24, L.5)

Recent studies have combined HD-MEA with the patch-clamp technique to simultaneously record the electrical activity of multiple single neurons or networks

References

Pasquale, V., Martinoia, S., and Chiappalone, M. (2017). Stimulation triggers endogenous activity patterns in cultured cortical networks. *Scientific Reports* 7. doi: 10.1038/s41598-017-08369-0

Lonardoni, D., Amin, H., Di Marco, S., Maccione, A., Berdondini, L., and Nieuwenhuis, T. (2017). Recurrently connected and localized neuronal communities initiate coordinated spontaneous activity in neuronal networks. *PLOS Computational Biology* 13, e1005672. doi: 10.1371/journal.pcbi.1005672

Reviewer #3 (Remarks to the Author):

Kobayashi et al. present an experimental framework that combines optical stimulation and HD-MEA recordings. The authors discuss the interesting problem of understanding to what extent single-neuron activity is related to network one, which is of possible interest for the journal reader. The authors apply the framework to cortical neurons cultured in monolayers.

Response: Thank you for appreciating the importance of the issues addressed in this paper and the promising results. We have corrected the points you raised and improved the quality of the paper. We are confident that the corrections have made the paper suitable for publication in *Nature Communications*.

The results are promising, but not as convincing. The cited literature in the introduction and discussion could be more elaborated and complete. The Matlab code seems to me to be not available. The description and illustration of the experimental setup in Figure 1 could be improved. The steps performed should be more clearly stated.

Response: Quantitative data have been added in the Results section to make it more convincing. We have also added references in the Introduction and Discussion sections. MATLAB code and sample data in a zip format have been attached. Additionally, the description of the experimental setup in Figure 1 has been improved. Specifically, the stimulus procedure and the steps performed in the light-stimulation experiment are also shown in Figure 1.

Quantitative data

Results (P.11, L.1)

In two independent samples, we stimulated 66 locations and detected 66 directly responding neurons and 125 indirectly responding neurons.

Results (P.13, L.5)

Of the indirectly responding neurons, 4% were neurons integrating two stimuli, and 0.8% were neurons responding to three stimuli (n = 130 neurons).

Results (P.14, L.13)

The top 36% of neurons (26 out of 73 neurons) showed a prolonged mean latency of >1 ms, with

92.7% of BDRC events being prolonged latencies. On the other hand, only 5.5% of neurons (4 out of 73 neurons) showed a decrease in BDRC, which tended to have a large standard deviation.

Results (P.15, L.11)

1.3% of neurons (4 locations out of 290 stimulation locations from eight independent samples) initiated network bursts when stimulated with a pulse of 0.2 Hz, with a network burst initiation probability of 10%, 18%, 60%, and 98%.

Experimental setup and steps for experiment

Results (P.9, L.5)

To determine the stimulation areas, fluorescence images of GFP were taken (Fig. 1d, step 1). The stimulation area was divided into grids and superimposed with the GFP fluorescence image (Fig. 1d, step 2). Grids containing ChR2-GFP-expressing neurons were manually selected (Fig. 1d, step 3). Optical stimulus was delivered to each selected locations sequentially (Fig. 1e).

Fig. 1

Fig.1 caption

(c) A photo of the recording system. (d) Procedure for selecting stimulation locations. A GFP fluorescence image were divided into grids. Stimulation locations were manually selected (red squares). (e) Optical stimulation procedure. Optical stimuli were delivered to each stimulation location sequentially.

Although the manuscript has some potential, it falls short at several points. e.g. it is not discussed how cell density and the experimental setup (e.g., volume of medium) might influence the performance of what they are proposing. This is the major limitation for the performance assessment. To me, the manuscript appears to be premature and at the current stage not suitable for Nature Communications.

Response: We have added to the Discussion section the conditions of the experiment, such as cell density and amount of medium, and the effect of changing the experimental conditions on the performance of the light stimulus. These modifications enabled us to properly evaluate performance and make it suitable for publication.

Discussion (P.22, L.18)

The results of this study depend on several experimental conditions. The first is the volume of culture medium: HD-MEA usually contains 800 μL of culture medium. The inner diameter of the ring that holds the culture medium is 19 mm, therefore the height of the medium is approximately 2.82 mm. The absorbance of Neurobasal medium for the light at a wavelength of 470 nm, which is used for optic stimulation, is approximately 0.5 at a path length of 1 cm^2 . Therefore, the intensity of the light may change when the volume of the culture medium is different. However, a 10% change in the culture medium changed the light intensity by 3.2%, indicating no significant change. Additionally, culture medium with low absorbance has been commercially available recent years, which will make optical stimulation experiments more robust with respect to the amount of culture media.

The second is the density of the cultured cells. Higher density results in better network bursts; however, it is difficult to distinguish individual neurons. At low densities, individual cells can be discriminated easily, but the number of surviving neurons is smaller, and network bursts are less likely to occur. In this study, the initial seeding density was fixed at 3000 cells/ mm^2 , but the density of cultured neurons at around one month in culture could vary depending on the number of times HD-MEA was reused and the viability of the cells at seeding. Therefore, optimizing culture conditions effectively stabilizes the experimental results.